# Evaluation of Effectiveness of Intervention Strategy in Control Simulation Experiment through Comparison with Model Predictive Control

Rikuto Nagai[1], Yang Bai[2], Masaki Ogura[2,1], Shunji Kotsuki[3,4], and Naoki Wakamiya[1]

[1]Graduate School of Information Science and Technology, Osaka University, Osaka, Japan
[2]Graduate School of Advanced Science and Engineering, Hiroshima University, Hiroshima, Japan
[3]Institute for Advanced Academic Research, Chiba University, Chiba, Japan
[4]Center for Environmental Remote Sensing, Chiba University, Chiba, Japan

**Correspondence:** Masaki Ogura (oguram@hiroshima-u.ac.jp)

**Abstract.** Climate change intensifies weather-related disasters, necessitating novel mitigation strategies beyond conventional weather prediction methods. The Control Simulation Experiment (CSE) framework proposes altering weather systems through small perturbations, but its effectiveness relative to other control methods remains uncertain. This study evaluates CSE's efficacy against Model Predictive Control (MPC), a well-established method in control engineering. We specifically develop an MPC algorithm tailored for the Lorenz-63 model, incorporating temporal deep unfolding to address challenges in controlling chaotic systems. Simulations show that MPC achieves higher success rates with less control effort under certain conditions, particularly with shorter prediction horizons. This work bridges control theory and atmospheric science, advancing the understanding of atmospheric controllability and informing future research efforts to mitigate extreme weather events.

## 1 Introduction

The Lorenz-63 system is a foundational model in the study of chaotic dynamics, originally developed to illustrate the unpredictable nature of atmospheric convection (Lorenz, 1963). Despite its simplicity, it captures essential features of deterministic chaos, such as sensitivity to initial conditions and the emergence of structured yet non-repeating trajectories. As a result, it has become a standard testbed for evaluating control methodologies in chaotic systems (Ott, 2002; Palmer, 1993).

Recent interest in weather control has further elevated the relevance of Lorenz-63, as atmospheric processes exhibit similar chaotic characteristics (Palmer, 2000). Weather systems are highly complex, characterized by high-dimensionality, partial observability, and chaotic behavior, making their control particularly challenging (Hoffman, 2002; Jarvis et al., 2008; Ban-Weiss and Caldeira, 2010). However, despite its simplified nature, the Lorenz-63 system remains a valuable tool for studying fundamental properties of chaotic systems (Palmer, 2000). Atmospheric processes exhibit similar chaotic characteristics, making Lorenz-63 an idealized yet insightful model for exploring weather control challenges, particularly in understanding how small perturbations influence trajectory evolution – a key aspect of weather modification. Moreover, the "wings" of the Lorenz attractor have often been used as a conceptual analogy for weather regime transitions (Weller and Schulz, 2014; Soldatenko,

2018). Studying control strategies within this simplified framework aids in technique development and lays the foundation to adapt to more complex models for weather control (Sierra et al., 2021; Weller and Schulz, 2014).

Weather control still remains underdeveloped due to both physical and mathematical challenges. From a physical perspective, atmospheric processes possess vast energy reserves, while human technological capabilities for intervention are orders of magnitude smaller. This discrepancy necessitates identifying physically justified methods for applying small perturbations to atmospheric variables, such as using sensitivity approaches in dynamical systems (Hall et al., 1982; Daniel J. Lea and Haine, 2000; Soldatenko and Chichkine, 2016). From a mathematical perspective, the atmospheric system and its components (including the atmosphere) are high-dimensional, chaotic, and only partially observed, making observability and controllability difficult to achieve. Additionally, defining realistic objective functions (cost functions) for control remains a challenge (Jarvis et al., 2009; Sun et al., 2023). These issues become even more critical when transitioning from conceptual models to more realistic ones (Weller and Schulz, 2014; Soldatenko, 2018), as seen in studies on closed-loop control of the global carbon-climate system (Sierra et al., 2021; Weller and Schulz, 2014). Given these challenges, it is essential to first study weather control in simpler systems that still exhibit chaotic characteristics. Although the Lorenz-63 system is a simplified model, it provides a controlled environment in which researchers can explore how small perturbations influence system evolution and test various control strategies.

However, effectively implementing control in chaotic systems requires experimental approaches that go beyond traditional predictability-focused studies. A promising approach in this context is the Control Simulation Experiment (CSE), introduced by Miyoshi and Sun (2022), which extends traditional numerical experiments by actively applying small perturbations to guide system trajectories, offering valuable insights into chaotic system control. The CSE approach leverages the system's inherent sensitivity to initial conditions, aiming to influence its long-term evolution with minimal interventions. The CSE extends the traditional Observing Systems Simulation Experiment (OSSE) (Atlas, 1997) by shifting the focus from improving predictability to exploring the potential for controlling chaotic systems (Miyoshi and Sun, 2022; Sun et al., 2023). Via infinitesimal perturbations, CSE aims to influence the future evolution of chaotic systems toward more desirable trajectories. This approach leverages the inherent sensitivity to initial conditions in chaotic systems, a phenomenon popularly known as the "butterfly effects" (Palmer, 1993; Ott, 2002), to steer the system's trajectory with minimal interventions.

While CSE has shown promise in specific contexts (Miyoshi and Sun, 2022; Sun et al., 2023), its performance relative to other established control methodologies has not been thoroughly examined (Kawasaki and Kotsuki, 2024). Hence, there is a need to compare the CSE with control strategies extensively studied and applied in other fields of engineering (Slotine and Li, 1991; Dorf and Bishop, 2011). Among them, Model Predictive Control (MPC) is one such method (see, e.g., Richalet et al., 1978; Mesbah, 2016), known for its robustness and ability to handle multi-variable control problems with explicit consideration of constraints and optimization objectives. The MPC predicts future system behavior using dynamic models and computes optimal control actions over a moving time horizon, making it a powerful tool for dynamical systems where future states are influenced by current interventions.

The objective of this study is to rigorously assess the efficacy of the CSE framework in comparison to MPC. By developing an MPC-based algorithm tailored for the Lorenz-63 model and integrating a refined version of MPC that incorporates deep

unfolding techniques (Kishida and Ogura, 2022; Hershey et al., 2014), we seek to explore whether established control methodologies can enhance the controllability of chaotic atmospheric models. This approach allows us to leverage MPC's strengths while accommodating the complexities of chaotic dynamics. To facilitate this investigation, we conduct comprehensive simulation experiments comparing the performance of the MPC-based control method with the strategies employed in the CSE. The results suggest that, when the control effort (instantaneous magnitude of control input applied at each time step) is limited and the prediction horizon is short, the MPC framework outperforms the CSE strategy in terms of control success rate. These improvements underscore the potential benefits of integrating well-established control methodologies into CSE-like frameworks for chaotic systems. This finding is particularly relevant in practical scenarios such as weather control, where strong interventions are physically infeasible, and effective regulation must therefore rely on minimal and gradual inputs. Moreover, since long-term forecasts are inherently unreliable, the prediction horizon should be kept short to ensure robust and timely control.

The contribution of this study is twofold. On the one hand, we enhance the traditional MPC approach with temporal deep unfolding to handle the unique challenges posed by chaotic systems, thereby extending MPC's applicability to nonlinear contexts. On the other hand, we provide a detailed comparative analysis between the MPC-based method and existing CSE strategies, advancing the understanding of controllability in chaotic dynamics and laying the groundwork for future research in this area. By comparing CSE and MPC within the same framework, our study seeks to evaluate the potential advantages and limitations of conventional control techniques in the domain of weather control applications. The insights gained from this comparison can inform the development of more effective control strategies and guide future research efforts aimed at mitigating the impact of extreme weather events through controllability.

We finally remark that, recently, Kawasaki and Kotsuki (2024) introduced MPC into the CSE framework and demonstrated its effectiveness in leading the Lorenz-63 system toward a prescribed regime. Their approach involves solving an optimal control problem by deriving and iteratively solving the necessary conditions for optimality using numerical methods. While this method shows promising results for low-dimensional systems like the Lorenz-63 model, it relies on analytical derivations and iterative computations that may not scale well to higher-dimensional or more complex systems due to increased computational demands. For this reason, we have chosen to use an MPC based on deep unfolding techniques, in which no symbolic execution is required.

The structure of the paper is as follows. Section 2 provides an overview of the CSE framework, including its theoretical foundations and methodologies. Section 3 details the development of the MPC method based on temporal deep unfolding and its implementation for the Lorenz-63 model. Section 4 presents the results of our comparative analysis between the MPC-based control method and the traditional CSE strategies. Finally, conclusions are drawn in Section 5.

## 2   Control Simulation Experiment

The CSE (Miyoshi and Sun, 2022) is a framework designed to explore the controllability of chaotic systems in the context of weather control applications. In this section, we provide a comprehensive description of the Lorenz-63 model (Lorenz, 1963),

the control target in CSE. We then present an overview of the control strategies employed within the CSE framework, detailing the methods used to influence the system's behavior. Finally, we discuss the limitations and challenges associated with these control strategies.

## 2.1  Control objective

The Lorenz-63 model is a simplified mathematical model that captures the essence of atmospheric convection and exhibits
chaotic behavior (Kravtsov and Tsonis, 2021). It has become a canonical example in the study of dynamical systems due to its sensitive dependence on initial conditions. The model consists of a set of three coupled, nonlinear ordinary differential equations representing the evolution of three state variables $x$, $y$, and $z$, which correspond to idealized atmospheric quantities.

The controlled Lorenz-63 model in the CSE framework incorporates control inputs $u_x$, $u_y$, and $u_z$ for each state variable, allowing for the application of external influences to the system:

$$\frac{dx}{dt} = \sigma(y - x) + u_x, \tag{1}$$

$$\frac{dy}{dt} = x(\rho - z) - y + u_y, \tag{2}$$

$$\frac{dz}{dt} = xy - \beta z + u_z. \tag{3}$$

In these equations, $\sigma$, $\rho$, and $\beta$ are positive parameters representing the Prandtl number, Rayleigh number, and a geometric factor, respectively. These parameters are set to $\sigma = 10$, $\rho = 28$, and $\beta = 8/3$, values known to produce chaotic dynamics
characterized by a butterfly attractor.

In the CSE framework, the Lorenz-63 model is discretized in time using the Runge-Kutta method with a step size of $0.01$. Under these discretization parameters, the system's solution is known to exhibit chaotic behavior, with the first state variable $x$ oscillating between positive and negative regions. This transition between regimes is referred to as a regime shift.

The primary control objective in CSE is to prevent regime shifts by maintaining the state variable $x$ within the positive
region. By achieving this, the system can be stabilized in a desired regime. Controlling such a chaotic system is challenging due to its inherent sensitivity to initial conditions and nonlinear dynamics, necessitating sophisticated control strategies.

## 2.2  Control strategies

The control objective is achieved by applying control inputs of a predetermined magnitude to each state variable $x$, $y$, and $z$ within the Lorenz-63 model to stabilize the system and avoid transitions into the negative regime.
To assess the effectiveness of this control strategy, a reference trajectory that represents the "true" system behavior is necessary. This reference, known as the Nature Run (NR), is generated by running a long-term simulation of the uncontrolled Lorenz-63 system. We denote the state of the NR at time $t$ as $\mathbf{x}_t^{\mathrm{NR}} \in \mathbb{R}^3$, where the three components correspond to the variables $x$, $y$, and $z$. Within the CSE framework, the NR serves as the ground truth. However, its true state variables are not directly observable; instead, noisy observations are obtained by adding independent Gaussian noise with a variance of $2.0$ to
each variable. To estimate the true values of the variables in the NR from these noisy observations, a data assimilation technique

known as the ensemble Kalman filter (Houtekamer and Zhang, 2016) is employed. The data assimilation interval is denoted as $T_a$. In the following analysis, we use $T_a = 8$ steps, consistent with the setup in CSE.

An overview of the control strategies proposed in CSE is provided below; for further details, refer to Miyoshi and Sun (2022).

1. Observation: An observation is obtained by adding Gaussian noise to the NR, which is generated by independently running the Lorenz-63 model with a fixed set of parameters initialized with a given initial condition.

2. Data Assimilation: At time step $t$, the ensemble Kalman filter assimilates the latest observations to update the estimate of the state of the NR. This provides the initial condition for subsequent forecasting.

3. Ensemble Forecasting: An ensemble of forecasts is generated from time $t$ to $t + T$ (where $T$ is the ensemble forecast horizon) using analysis ensembles, which are slightly perturbed initial conditions. These perturbations simulate uncertainties in the initial state and model errors. Typically, three ensemble members are generated to estimate the future state of the NR.

4. Regime Shift Detection: The ensemble forecasts are analyzed to determine if any member predicts a regime shift (i.e., the state variable $x$ crossing into the negative region) within the forecast horizon. If at least one ensemble member indicates a regime shift, control actions are deemed necessary.

5. Control Input Determination: For each time step from $t + 1$ to $t + T_a - 1$, control inputs $u_x$, $u_y$, and $u_z$ are calculated to prevent the predicted regime shift. The control inputs are determined by:

   (a) Selecting an ensemble member $S$ that predicts a regime shift and an ensemble member $N$ that does not.

   (b) Computing the difference $S - N$ between the two ensemble members at each time step.

   (c) Normalizing this difference vector so that its Euclidean norm equals a predetermined magnitude $D$.

   (d) Applying the normalized control inputs to the NR at each time step.

6. Iteration: After applying the control inputs, new observations are obtained at time $t + T_a$, and the process repeats from step 1.

We adopt the above method as a baseline for comparison with other methods in the following tests. To ensure a fair evaluation, we follow the original setting described in their work.

## 2.3   Limitations of CSE

Despite the promising results achieved by the control strategies within the CSE framework, several limitations impact their overall effectiveness in controlling chaotic systems like the Lorenz-63 model. One significant limitation is the sensitivity of the control strategy to the ensemble forecast horizon $T$. Experimental observations have indicated that when the prediction horizon is relatively long, the success rate of preventing regime shifts is high. This is because a longer prediction horizon

allows for earlier detection of potential regime shifts, providing sufficient lead time to apply control inputs effectively. However, extending the prediction horizon also increases computational costs and introduces greater uncertainty due to the chaotic nature of the system. Small errors in the initial conditions can grow exponentially over time, leading to significant deviations in the forecasted trajectories. Conversely, when the prediction horizon is relatively short, the control strategy's success rate decreases significantly. Shorter horizons may not provide enough time to detect and counteract impending regime shifts, resulting in a higher likelihood of the system transitioning into undesirable states.

Another challenge lies in determining the appropriate magnitude of the control inputs, represented by the Euclidean norm $D$ of the control vector $\mathbf{u} = [u_x, u_y, u_z]^\top$. If $D$ is set too small, the control inputs may be insufficient to influence the system's dynamics and prevent regime shifts. On the other hand, if $D$ is too large, the control actions may be impractical for implementations due to physical limitations. Balancing the magnitude of the control inputs to achieve effective control without causing adverse side effects is a delicate task. This issue underscores the need for a more systematic approach to determine optimal control input magnitudes that consider both effectiveness and feasibility.

Additionally, the CSE framework lacks an optimization mechanism capable of addressing the multiple objectives and constraints inherent in practical control applications. Real-world control strategies must often balance competing goals, such as minimizing control effort, satisfying physical constraints, and achieving desired system performance. Without such a mechanism, the CSE approach may produce suboptimal control inputs, limiting its efficiency, practicality, and applicability to complex systems or real-world scenarios where these considerations are critical.

These limitations highlight the necessity for alternative or enhanced control methodologies capable of addressing the inherent challenges of controlling chaotic systems effectively. In particular, there is a need for control strategies that can optimize control inputs while explicitly considering system constraints, uncertainties, and multiple objectives. Control methods such as MPC offer a promising avenue in this regard. The MPC provides a systematic framework for optimizing control actions over a future horizon while handling multivariable systems with explicit constraints and objectives. By integrating such methods, it may be possible to overcome the limitations of the current CSE approach, enhancing the controllability of chaotic systems like the Lorenz-63 model and improving the practical feasibility of weather control applications.

## 3 MPC Using Temporal Deep Unfolding

MPC (Richalet et al., 1978; Mesbah, 2016) is a prominent control strategy extensively used in engineering disciplines for its capability to handle multivariable control problems with constraints and to anticipate future system behavior by solving an optimization problem at each time step. Unlike the CSE strategy, which employs an event-triggered scheme and introduces perturbations only when a regime change is detected in the ensemble, the MPC framework applies continuous perturbations to the nature run at every time step.

In this study, we compare the effectiveness of MPC with that of the CSE in terms of control success rates for chaotic systems like the Lorenz-63 model. Among the various MPC methodologies, we adopt an advanced version known as MPC using temporal deep unfolding, which has demonstrated effectiveness in controlling nonlinear systems (see, e.g., Kishida and

Ogura, 2022; Liu et al., 2024; Aizawa et al., 2024). This section provides a comprehensive overview of MPC and elaborates on how temporal deep unfolding enhances its capabilities, particularly in the context of controlling chaotic dynamics.

## 3.1 MPC

MPC is a control strategy that optimizes control inputs by predicting future states of a dynamic system over a finite prediction horizon at each discrete time step (Rawlings and Mayne, 2009). By solving an optimization problem that minimizes a predefined cost function, MPC adjusts the control inputs to ensure that the system's output follows a desired trajectory while satisfying constraints on inputs and states. The key components of MPC include

- System Model: A mathematical representation of the system dynamics, which can be linear or nonlinear, deterministic or stochastic;

- Cost Function: An objective function that quantifies the performance of the system, incorporating terms for tracking error, control effort, and possibly other considerations like energy consumption or economic costs;

- Constraints: Physical or operational limitations on the control inputs and system states, such as actuator limits, safety requirements, or environmental regulations.

The discrete-time state equation governing the controlled system is expressed as

$$\mathbf{x}_{t+1} = f(\mathbf{x}_t, \mathbf{u}_t), \tag{4}$$

where $\mathbf{x}_t \in \mathbb{R}^3$ is the system state and $\mathbf{u}_t \in \mathbb{R}^3$ is the control input. The objective is to find a sequence of control inputs $\{\mathbf{u}_t, \mathbf{u}_{t+1}, \ldots, \mathbf{u}_{t+T-1}\}$ over a prediction horizon $T$ that minimizes a cost function $J$ while satisfying the system dynamics and constraints. This formulation of $J$ enables the controller to minimize the cumulative cost $J$ while adhering to dynamic constraints and specified limits on control inputs and states. By iteratively solving this optimization problem, MPC adapts to changing dynamics and disturbances, enhancing system performance.

The optimization problem at each time step can be typically formalized as follows:

$$
\begin{aligned}
\underset{\{\mathbf{u}_{t+k}\}_{k=0}^{T-1}}{\text{minimize}} \quad & J \\
\text{subject to} \quad & \mathbf{x}_{t+k+1} = f(\mathbf{x}_{t+k}, \mathbf{u}_{t+k}), \quad k = 0, \ldots, T-1, \\
& \mathbf{u}_{\min} \leq \mathbf{u}_{t+k} \leq \mathbf{u}_{\max}, \quad k = 0, \ldots, T-1, \\
& \mathbf{x}_{\min} \leq \mathbf{x}_{t+k} \leq \mathbf{x}_{\max}, \quad k = 0, \ldots, T
\end{aligned}
\tag{5}
$$

where $\mathbf{u}_{\min}$ and $\mathbf{u}_{\max}$ represent input constraints while $\mathbf{x}_{\min}$ and $\mathbf{x}_{\max}$ represent state constraints.

This formulation enables the controller to minimize the cumulative cost $J$ while adhering to the system dynamics and specified constraints. By iteratively solving this optimization problem at each time step and implementing only the first control input $\mathbf{u}_t$, MPC adapts to changing dynamics and disturbances, enhancing system performance and robustness.

## 3.2 Temporal deep unfolding

Deep unfolding is a methodology that bridges the gap between iterative optimization algorithms and deep learning architectures (Hershey et al., 2014; Jagannath et al., 2021). It involves unfolding an iterative algorithm into a layer-wise structure resembling a neural network, where each iteration corresponds to a layer. Parameters within the algorithm can then be learned using backpropagation and gradient-based optimization techniques.

Temporal deep unfolding extends this concept to dynamic systems by treating the state evolution equations as an iterative algorithm (Kishida and Ogura, 2022; Aizawa et al., 2024). In this approach, the system dynamics over a prediction horizon $T$ are unfolded into a feed-forward network with $T$ layers, each representing the system's state at a future time step. The control inputs $\{\mathbf{u}_t, \mathbf{u}_{t+1}, \ldots, \mathbf{u}_{t+T-1}\}$ are treated as learnable parameters within this network. By leveraging deep learning techniques such as backpropagation, the control inputs are optimized to minimize the cost function. This approach offers several advantages. First, in terms of efficiency, gradient-based optimization can be more efficient than traditional optimization methods, especially for large-scale or complex problems. Second, it provides flexibility by handling nonlinear dynamics and cost functions. Additionally, it facilitates the integration with machine learning, allowing for the incorporation of learning-based components, such as neural network approximations of dynamics or cost functions.

In the context of temporal deep unfolding, incremental learning is a technique where the model learns from new data incrementally without retraining from scratch, preserving knowledge from previous learning. Specifically, incremental learning involves progressively increasing the prediction horizon $T$. In this study, we employ incremental learning to refine the control inputs obtained through temporal deep unfolding (denoted as MPCIL), enhancing the MPC's ability to control the Lorenz-63 model effectively.

## 3.3 Proposed control algorithm

The control system implemented in this study is controlled by adding the control inputs obtained by MPC using temporal deep unfolding to the three variables of the Lorenz-63 model at each time step. The algorithm proceeds as follows.

1. Observation: An observation, $\mathbf{x}_t$, is obtained by adding Gaussian noise to the NR $\mathbf{x}_t^{\mathrm{NR}}$.

2. Temporal deep unfolding: Using the observation, control inputs $\{\mathbf{u}_t, \mathbf{u}_{t+1}, \ldots, \mathbf{u}_{t+T_a-1}\}$ are determined as follows.

   (a) We construct a feed-forward network with $T$ layers. Each layer represents the state transition from time $t+k$ to time $t+k+1$ ($k = 0, 1, \ldots, T-1$) and incorporates the control input $\mathbf{u}_{t+k}$.

   (b) The sequence of control inputs is then initialized, which can be done using previous control inputs or random values.

   (c) Using backpropagation and gradient descent, the control inputs $\{\mathbf{u}_t, \mathbf{u}_{t+1}, \ldots, \mathbf{u}_{t+T-1}\}$ are updated to minimize the cost function

$$J_{t,T} = \sum_{i=0}^{T} c(-\mathbf{x}_t), \tag{6}$$

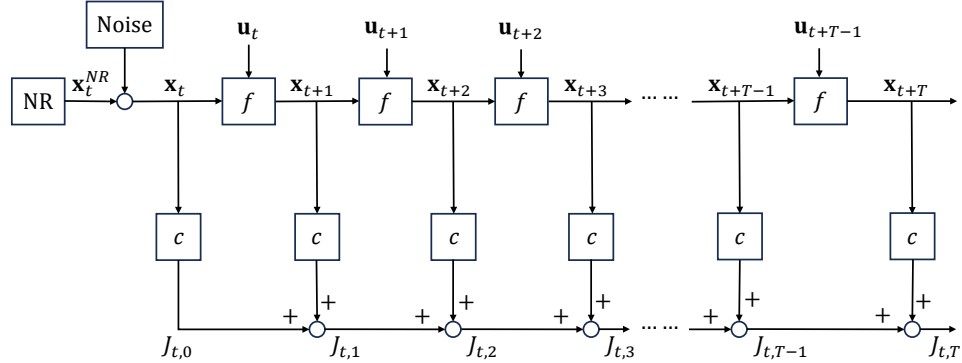

**Figure 1.** Conceptual illustration of model predictive control via temporal deep unfolding

where

$$c(\mathbf{x}) = \begin{cases} 0, & \text{if } x < 0, \\ x, & \text{otherwise}, \end{cases} \tag{7}$$

penalizes negative values of $x$, the first component of the vector state variable $\mathbf{x}$. To ensure consistency with the CSE control strategy in which the magnitude of the control input is normalized to $D$, the MPC formulation trains control inputs whose magnitude is also constrained to $D$.

3. Iteration: After applying the control inputs $\{\mathbf{u}_t, \mathbf{u}_{t+1}, \ldots, \mathbf{u}_{t+T_a-1}\}$ to the NR, new observations are performed at time $t + T_a$, and the process repeats from step 1.

When incremental learning is applied, Step 2c becomes as follows.

(c') Using backpropagation and gradient descent, the control inputs $\{\mathbf{u}_t, \mathbf{u}_{t+1}, \ldots, \mathbf{u}_{t+T-1}\}$ are updated to minimize the cost functions $J_{t,0}, J_{t,1}, \ldots, J_{t,T}$ sequentially.

The algorithm above can be illustrated more straightforwardly by Figure 1.

## 4 Evaluation Results

In this section, we evaluate the effectiveness of the control strategy employed in the CSE by comparing it with MPC using temporal deep unfolding applied to the Lorenz-63 model. Miyoshi and Sun (2022) conducted experiments performing CSE for 40 different initial conditions (each initial condition corresponds to a different NR) and investigated the control success rate. We adopted the results from Miyoshi and Sun (2022) as a baseline to avoid discrepancies that might arise from re-implementing the CSE scheme independently. This ensures that observed performance differences are due to the control strategies themselves, not implementation variability. By comparing their results with the control success rate obtained using MPC with temporal deep unfolding, we aim to assess the effectiveness of the control policies employed in CSE.

**Table 1.** Parameters of the control system

| Lorenz-63 model | |
| --- | --- |
| $\sigma$ | 10 |
| $\rho$ | 28 |
| $\beta$ | 8/3 |
| **Parameters common in CSE, MPC, and MPCIL** | |
| Step size | 0.01 |
| Upper limit of the Euclidean norm of the control input | $D$ |
| Length of NRs | 8,000 |
| Prediction horizon | $T$ |
| Observation period | $T_a = 8$ |
| **Parameters common in MPC and MPCIL** | |
| Optimizer | Adam |
| Learning rate | 10 |
| The number of training iterations using error backpropagation | $m$ |

## 4.1 Parameters

We detail the parameters used in the control system of the Lorenz-63 model when applying MPC with temporal deep unfolding in Table 1. As stated in Section 2.1, standard parameters of the Lorenz-63 model are set as $\sigma = 10$, $\rho = 28$, $\beta = 8/3$. Each of the component $u_x$, $u_y$, and $u_z$ of the initial control inputs before training are initialized as random numbers drawn from a normal distribution with a mean of 0 and a standard deviation of 1. These are then normalized in such a way that the vector $u_t$ has the predetermined Euclidean norm of $D$ . The learning rate, which determines the step size during the optimization process, is set to 10. The number of training iterations using error backpropagation is denoted as $m$. We utilize the Adam optimizer (Bae et al., 2019) , a widely used optimization algorithm in deep learning, to minimize the cost function. The target value for the cost function is set to 0, allowing the control inputs at each step to be learned so that the cost function approaches this target. Control is performed for various combinations of the prediction horizon $T$ (number of steps ahead in the prediction) and the upper limit $D$ of the Euclidean norm of the control input vector.

## 4.2 Control results of the Lorenz-63 model using MPC

In this section, we present the results of controlling the Lorenz-63 model using MPC with temporal deep unfolding. The initial values of the system were set according to Miyoshi and Sun (2022) as $(x, y, z) = (8.20747939, 10.0860429, 23.86324441)$. The values of the control inputs, $x$-coordinate, and the values of the variables $x$, $y$, and $z$ for each combination of parameters $T$ and $D$ are shown in Figure 2. Additionally, in Figure 3, we illustrate how the control inputs $u_x$, $u_y$, and $u_z$ evolve with the number of parameter updates $m$. This provides insight into how the inputs are adjusted as learning progresses. These figures

**Table 2.** Comparison of control success rates between CSE and MPC, where $P_{\text{CSE}}$ and $P_{\text{MPC}}$ represent their respective success rates.

| | $T = 113$ | | $T = 151$ | | $T = 188$ | | $T = 226$ | | $T = 301$ | |
|---|---|---|---|---|---|---|---|---|---|---|
| | $P_{\text{CSE}}$ | $P_{\text{MPC}}$ | $P_{\text{CSE}}$ | $P_{\text{MPC}}$ | $P_{\text{CSE}}$ | $P_{\text{MPC}}$ | $P_{\text{CSE}}$ | $P_{\text{MPC}}$ | $P_{\text{CSE}}$ | $P_{\text{MPC}}$ |
| $D = 0.02$ | 0.000 | 0.000 | 0.000 | 0.000 | 0.000 | 0.000 | 0.000 | 0.000 | 0.050 | 0.000 |
| $D = 0.03$ | 0.000 | 0.000 | 0.000 | 0.000 | 0.000 | 0.000 | 0.050 | 0.000 | 0.975 | 0.000 |
| $D = 0.04$ | 0.000 | 0.000 | 0.000 | 0.000 | 0.000 | 0.000 | 0.425 | 0.000 | 0.975 | 0.025 |
| $D = 0.05$ | 0.000 | 0.000 | 0.000 | 0.000 | 0.025 | 0.025 | 0.550 | 0.075 | 0.975 | 0.150 |
| $D = 0.1$ | 0.000 | 0.400 | 0.000 | 0.375 | 0.250 | 0.375 | 0.800 | 0.575 | 0.825 | 0.525 |
| $D = 0.2$ | 0.000 | 0.725 | 0.025 | 0.600 | 0.275 | 0.800 | 0.675 | 0.800 | 0.825 | 0.725 |
| $D = 0.3$ | 0.000 | 0.750 | 0.000 | 0.725 | 0.250 | 0.850 | 0.400 | 0.750 | 0.725 | 0.650 |
| $D = 0.4$ | 0.000 | 0.650 | 0.000 | 0.750 | 0.200 | 0.725 | 0.100 | 0.675 | 0.500 | 0.400 |
| $D = 0.5$ | 0.025 | 0.600 | 0.000 | 0.625 | 0.175 | 0.400 | 0.200 | 0.125 | 0.525 | 0.025 |

**Table 3.** Comparison of control success rates between MPCIL and MPC, where $P_{\text{MPCIL}}$ and $P_{\text{MPC}}$ represent their respective success rates.

| | $T = 19$ | | $T = 38$ | | $T = 57$ | |
|---|---|---|---|---|---|---|
| | $P_{\text{MPCIL}}$ | $P_{\text{MPC}}$ | $P_{\text{MPCIL}}$ | $P_{\text{MPC}}$ | $P_{\text{MPCIL}}$ | $P_{\text{MPC}}$ |
| $D = 0.02$ | 0.000 | 0.000 | 0.000 | 0.000 | 0.000 | 0.000 |
| $D = 0.03$ | 0.000 | 0.000 | 0.000 | 0.000 | 0.000 | 0.000 |
| $D = 0.04$ | 0.000 | 0.000 | 0.000 | 0.000 | 0.000 | 0.000 |
| $D = 0.05$ | 0.000 | 0.000 | 0.000 | 0.000 | 0.000 | 0.000 |
| $D = 0.1$ | 0.000 | 0.000 | 0.675 | 0.000 | 0.725 | 0.675 |
| $D = 0.2$ | 0.825 | 0.000 | 0.900 | 0.900 | 0.900 | 0.875 |
| $D = 0.3$ | 0.975 | 0.975 | 0.925 | 0.925 | 0.975 | 0.975 |
| $D = 0.4$ | 1.000 | 1.000 | 1.000 | 1.000 | 1.000 | 1.000 |
| $D = 0.5$ | 1.000 | 1.000 | 1.000 | 1.000 | 1.000 | 1.000 |

correspond to the first time step of the MPC implementation, where the control input at time step 0 is optimized while predicting the system's behavior over a prediction horizon of $T = 113$ steps with a control limit of $D = 0.5$.

We conducted control experiments with the prediction horizon $T = 113$ and various values of the upper limit $D$ of the Euclidean norm of the control input, specifically $D = 0.5$, 0.4, 0.3, and 0.2. Figure 2 illustrates the time series of the control inputs, the $x$-coordinate values, and the trajectories of the system for each combination of $T$ and $D$. The left column shows how the control inputs evolve over time, the middle column depicts the time series of the $x$-coordinate, and the right column presents the trajectory of the system in the phase space.

The illustrative example above demonstrates the effectiveness of our control method. To further assess its feasibility, we conducted systematic evaluations using a broader set of initial conditions. Specifically, we conducted simulations for 40 different initial conditions to examine the control success rate, defined as the proportion of successful control instances. Following the methodology in Miyoshi and Sun (2022), we varied the prediction horizon $T$ and the upper limit $D$ of the Euclidean norm of the control input. Additionally, to ensure consistent experimental conditions, the 40 initial values used in this study are the same as those used by the original CSE. Table 2 presents a comparison of the control success rates between CSE and our MPC approach for different values of $T$ and $D$. We also investigated the effect of incremental learning when performing control us-

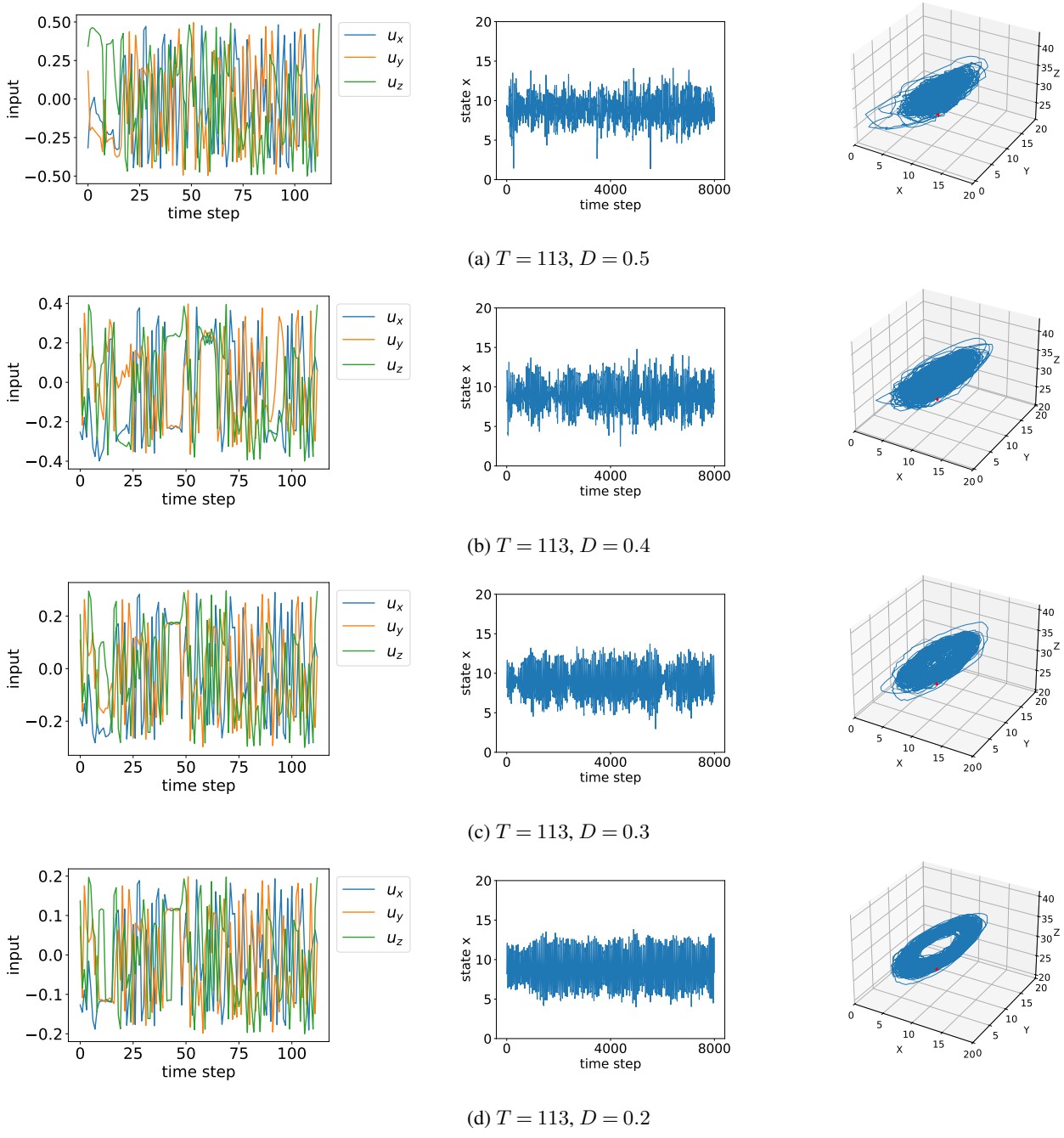

(a) $T = 113, D = 0.5$

(b) $T = 113, D = 0.4$

(c) $T = 113, D = 0.3$

(d) $T = 113, D = 0.2$

**Figure 2.** Control of the Lorenz-63 model using MPC. Left figure: time series of control inputs, central figure: time series of $x$ coordinate values, right figure: trajectory of the system.

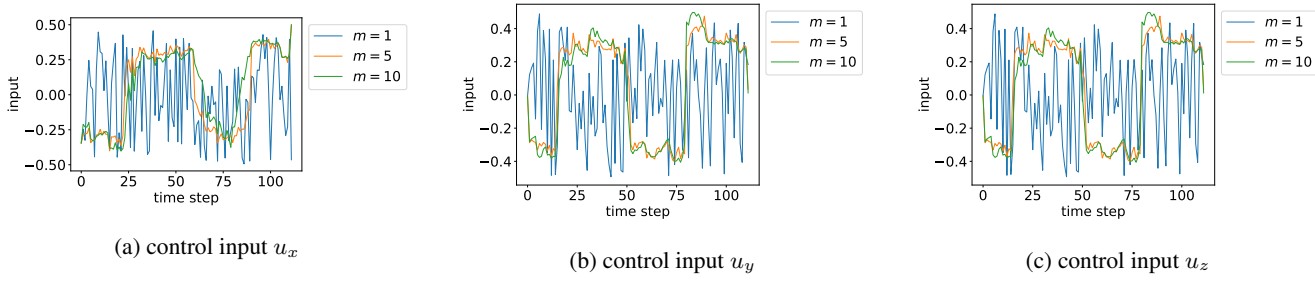

(a) control input $u_x$

(b) control input $u_y$

(c) control input $u_z$

**Figure 3.** The process of training control inputs at the initial time instant when $T = 113$ and $D = 0.5$. Here, $m$ denotes the number of training iterations during optimization.

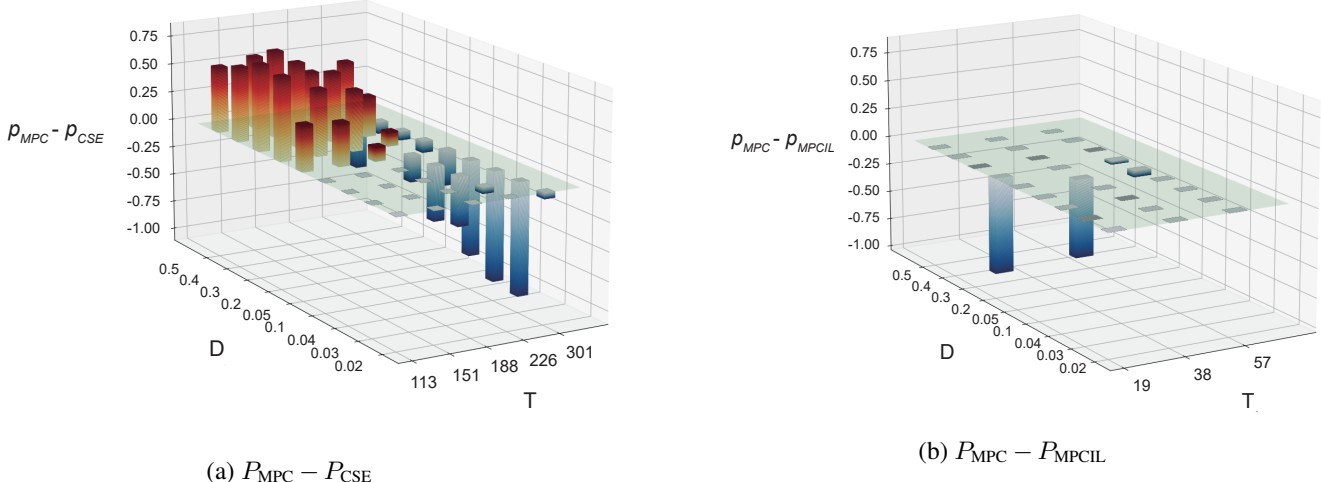

(a) $P_{\mathrm{MPC}} - P_{\mathrm{CSE}}$

(b) $P_{\mathrm{MPC}} - P_{\mathrm{MPCIL}}$

**Figure 4.** Comparison of control success rates: (a) between CSE and MPC, where $P_{\mathrm{MPC}} - P_{\mathrm{CSE}}$ represents the difference in success rates, and (b) between MPCIL and MPC, where $P_{\mathrm{MPC}} - P_{\mathrm{MPCIL}}$ denotes their success rate difference. Red bars indicate cases where MPC outperforms, while blue bars show where MPC is less effective.

ing MPC. Table 3 compares the control success rates obtained using MPC without incremental learning and with incremental learning (denoted as MPCIL).

A detailed analysis of these results is provided in the the next section. In particular, we discuss the differences in control strategies between MPC and CSE, the impact of prediction horizon length and control input limits on performance, and the trade-offs between control accuracy and computational efficiency. These insights contribute to a better understanding of how advanced control methodologies can be effectively applied to complex atmospheric models.

### 4.3 Discussion

Figure 4 shows the comparison of the control success rate for each control method (MPC, CSE, and MPCIL) based on the results of Table 2. The comparison between MPC and CSE is illustrated by Figure 4.a, and that between MPC and MPCIL is illustrated by Figure 4.b.

Based on the results from Figure 4.a, it is observed the larger $D$ is and the smaller $T$ is, the better the MPC performs. Specifically, the control using MPC performs better than control using CSE for $D$ values of 0.2 or higher. For $D = 0.1$, MPC shows better control performance for $T \leq 226$, whereas CSE exhibits higher control performance for $T \geq 226$. For $D \leq 0.05$, both MPC and CSE achieve 0% control success rate for $T \leq 188$, and CSE demonstrates better control performance for $T \geq 188$. This indicates that while CSE control is effective for longer prediction horizons, it is less effective for shorter prediction horizons.

In relatively short prediction horizons, the difference in control strategies between CSE and MPC could be considered as a cause for the lower success rate of control by CSE compared to MPC. In the control strategy of CSE, at each time step, the difference between ensemble members indicating regime shifts, denoted as $S$, and ensemble members not indicating regime shifts, denoted as $N$, is taken, and the control input is calculated based on that difference $S - N$. However, it is not necessarily the case that the value calculated from the difference of ensemble members always contributes to keeping the state of the Lorenz-63 model in the positive region with respect to the value of $x$. On the other hand, in control by MPC, a cost function is set up so that the cost increases when the value of $x$ is negative, causing the control input to be updated to make the value of $x$ positive.

Next, we will discuss the differences between using incremental learning and not using it in a control system under MPC. As seen in Figure 4.b, significant differences can be seen for $D = 0.2$ and $D = 0.1$. For $D = 0.2$ and $T = 19$, the control success rate was 0% without incremental learning but increased to 82.5% with incremental learning. For $D = 0.1$, $T = 38$, the control success rate with incremental learning was 67.5% higher than without incremental learning. The control success rate with incremental learning tended to be better than without it across the tested combinations of parameters $T$ and $D$ in Table 3.

On the other hand, we need to consider the impact of incremental learning on computation time. The average computation time of MPC-based control is 378s, 1272s, and 1965s for $T = 19$, $T = 38$, and $T = 57$, respectively. It increases almost linearly with the prediction horizon $T$ when control is performed without incremental learning. On the other hand, with incremental learning, the computation time is 16004s, 49556s, and 121578s. It can be observed that computation time increases significantly as the prediction horizon $T$ increases. This result indicates that MPCIL outperforms MPC in some cases but it requires a substantial amount of computational resources.

These findings highlight a trade-off between control performance and computational efficiency. In applications where computational resources are limited or real-time control is required, the standard MPC approach may be more practical despite its lower success rate in certain scenarios. Conversely, in situations where higher control accuracy is paramount and computational resources are ample, employing incremental learning with MPC can provide superior performance.

Overall, the MPC approach, especially when enhanced with incremental learning, demonstrates a strong potential for controlling chaotic systems like the Lorenz-63 model, outperforming the traditional CSE strategy under certain conditions. This suggests that advanced control methodologies from control engineering can be effectively applied to complex atmospheric models, potentially contributing to the development of more effective weather control strategies.

## 5    Summary and Future Work

In this study, we conducted a comprehensive evaluation of the control strategy employed in the CSE by comparing it with MPC using temporal deep unfolding on the Lorenz-63 model. The findings reported by Miyoshi and Sun (2022) indicate that the CSE strategy performs relatively well under longer prediction horizons, whereas its success rate diminishes when the horizon is short. In contrast, our results show that MPC achieves better control performance under short prediction horizons, maintaining the system's state within the desired regime.

This study contributes in two key ways: First, it presents an adaptation and tailored implementation of temporal deep unfolding for controlling the Lorenz-63 system within a control framework inspired by the CSE strategy. Second, it provided a detailed comparison between MPC-based methods and existing CSE strategies, advancing the understanding of controllability for chaotic dynamics. While this study primarily focuses on theoretical and numerical analyses, it broadens the applicability of MPC by demonstrating its effectiveness in addressing practical challenges associated with chaotic dynamics. One promising

avenue is the management of extreme weather events, where effectively understanding and controlling chaotic behavior is critical.

While our work provides a potential pathway toward weather control, several challenges remain for practical implementation due to the inherent simplicity of the Lorenz system. The classical Lorenz system, though widely used to study deterministic chaos and to test new numerical algorithms (Wang, 2013; Soldatenko and Chichkine, 2016), only captures the essential features

of chaotic dynamics. Moreover, the concept of a "weather regime" is itself an open problem, and equating weather regimes to the wings of the Lorenz-63 attractor remains a conceptual simplification. Therefore, more realistic models, including the coupled (fast-slow) versions of the Lorenz system motivated by the atmosphere-ocean interaction (Peña and Kalnay, 2004; Siqueira and Kirtman, 2012), are necessary to better address the challenges in the context of weather control (e.g., Ban-Weiss and Caldeira, 2010; Soldatenko, 2018). Furthermore, additional challenges such as physical feasibility and the formulation

of appropriate cost functions must be overcome to generalize the aforementioned conceptual insights into practical weather control strategies. These aspects will be explored in our future research.

*Code availability.*    The code that supports the findings of this study is available at https://github.com/ogura-lab/NPG_CSE-MPC.

*Data availability.* The authors declare that all data supporting the findings of this study are available within the figures and tables of the paper.

*Author contributions.* RN and MO conceptualized this study. RN conducted the numerical experiments. RN, MO, and YB wrote the manuscript. MO, SK, and NW supervised and directed this study.

*Competing interests.* The authors declare that they have no conflict of interest.

*Acknowledgements.* This study was partly supported by JST Moonshot R&D Grant Numbers JPMJMS2284 and JPMJMS2389, JSPS KAK-ENHI Grant Numbers JP21H04571 and JP22H00514, and the IAAR Research Support Program of Chiba University. The authors thank
program members of the Moonshot JPMJMS2284 for valuable discussion.

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
