# Peer review of "Evaluation of Effectiveness of Intervention Strategy in Control Simulation Experiment through Comparison with Model Predictive Control"

_Nonlinear Processes in Geophysics, 2024_

## Author Comment (AC2)

**Response to Anonymous Referee 1**

1. *The application of control theory, including optimal control theory, to solving problems of weather modification and climate engineering is not new, but at the same time, this problem remains underdeveloped. The application of control theory in meteorological and climate applications has two important aspects.*

   *The first aspect, let's call it physical, is related to the fact that atmospheric processes possess enormous energy, while human technological energy capabilities are orders of magnitude smaller. Therefore, the identification of physically justified methods of affecting weather and climate is a topical issue. For example, this problem can be solved using the sensitivity theory of dynamical systems [1-3].*

   *The second aspect concerns the mathematical theory of weather and climate control. Here too, rather complex problems arise related to the observability and controllability of dynamical systems, which the climate system and its components, including the atmosphere, are. An important and rather complex issue is also the problem of goal-setting, i.e., justification of objective functions (also known as cost functions ot performance indices).*

   **Response:** We appreciate the referee's concise summary of the two important aspects involved in applying control theory to meteorological and climate applications. This guidance has helped us strengthen our Introduction to address both the immense energy scale of atmospheric processes and the complexity of designing appropriate objectives for control.

2. *Unfortunately, the key problems of weather and climate control listed above are not even identified by the authors of the reviewed article. I see the reason for this in that the authors seem to be little familiar with a number of works in which the problem of weather and climate control was considered with varying degrees of detail. The list of some articles includes [4-11].*

   **Response:** We acknowledge the referee's comment that our original manuscript did not sufficiently highlight the fundamental challenges of weather climate control, nor reference the broader literature on this topic. In the revised manuscript, we have significantly revised the Introduction (the second paragraph) to explicitly discuss the key issues and to cite relevant previous works that the referee kindly suggested. We hope that these updates clarify how our approach fits within existing research and emphasize the main difficulties that must be addressed toward weather and climate control.

   **Revised Parts:** The following part, highlighting the challenges of weather control, was added to the Introduction section from line 24 of the revised manuscript.

   *"Nevertheless, weather control remains underdeveloped due to both physical and mathematical challenges ... "*

3. *As a model of the atmospheric system, the authors use the well-known E. Lorenz system, which under certain conditions exhibits chaotic behavior. However, it should be kept in mind that the sensitivity and controllability of chaotic dynamical systems have their distinctive features [12,13]. The Lorenz system, in its classical version, as well as in the coupled (fast-slow) version to mimic the atmosphere-ocean system [14,15], is widely used in many fields, including meteorology. This system is mainly used to study the phenomenon of deterministic chaos, as well as to test various new numerical*

*algorithms including AI. However to study the possibilities of controlling meteorological and climatic processes, models must be much more realistic (see for example [7,9]).*

**Response:** We thank the referee for kindly providing a valuable perspective regarding the use of the E. Lorenz system in the context of controlling meteorological and climatic processes. While we acknowledge that models for practical weather and climate control must be much more realistic than the classical Lorenz system, our primary objective was to provide a detailed comparative analysis between the MPC-based method and existing CSE strategies, thereby advancing the understanding of controllability in chaotic dynamics. Given that the CSE framework was originally introduced and tested using the E. Lorenz system, we believe that our research design is appropriate for establishing a conceptual foundation. Nevertheless, we have now explicitly addressed these limitations in the revised Introduction by discussing the distinctive features of chaotic dynamical systems (Wang, 2013; Soldatenko and Chichkine, 2016) and by referencing more realistic model approaches (Ban-Weiss and Caldeira, 2010; Soldatenko, 2018). We also highlight the challenges related to physical feasibility and cost-function design (Jarvis et al., 2009; Soldatenko and Yusupov, 2021) as crucial issues for future research in this area.

**Revised Parts:** The following part, elaborating on why Lorenz-63 was chosen, was added to the Introduction section from line 14 of the revised manuscript.

*" Recent interest in weather control has further elevated the relevance of Lorenz-63, as atmospheric processes exhibit similar chaotic characteristics. Weather systems are highly complex, characterized by high-dimensionality, partial ob-15 servability, and chaotic behavior, making their control particularly challenging. However, despite its simplified nature, the Lorenz-63 system remains a valuable tool for studying fundamental properties of chaotic systems ... "*

4. *The authors essentially solved the problem of studying the sensitivity of the Lorenz system to small controlled perturbations. Meanwhile they did not indicate the problems that may arise in this case [12,13]. The authors also did not at all illuminate the current state of the problem and the current achievements of other researchers, referring only to the work of Miyoshi and Sun. Despite the fact that the work is well written and structured, the authors need to substantially revise it taking into account the above mentioned comments.*

**Response:** We thank the referee for kindly summarizing the key points. In our original manuscript, we essentially solved the problem of studying the sensitivity of the Lorenz system to small controlled perturbations but, as the referee noted, we did not indicate the potential problems that may arise in this case, nor did we adequately illuminate the current state of the problem or the current achievements of other researchers beyond the work of Miyoshi and Sun. In response, we have substantially revised the Introduction section to explicitly identify these key problems and to incorporate a broader discussion of the literature, including the works cited in references (Wang, 2013; Soldatenko and Chichkine, 2016) and others. We hope that these revisions adequately address the referee's concerns and that the paper is now ready for publication.

**Revised Parts:** A broader discussion of the literature was added to the Conclusion section of the revised manuscript.

*" While our work provides a potential pathway toward weather control, several challenges remain for practical implementation350 due to the inherent simplicity of the Lorenz system. The classical Lorenz system, ... "*

5. *References*

*[1] Hall M.C.G., Cacuci D.G. Sensitivity analysis of a radiative-convective model by the adjoint method. J. Atmos. Sci. 1982, 39, 2038-2050.*

*[2] Lea D., Allen M., Haine T. Sensitivity analysis of the climate of a chaotic system. Tellus, 2000, 52A, 523–532.*

*[3] Soldatenko S., Yusupov R. The determination of feasible control variables for geoengineering and weather modification based on the theory of sensitivity in dynamical systems. Journal of Control Science in Engineering, 2016, 2016, 1547462*

*[4] Hoffman, R.N. Controlling the global weather. Bull. Am. Meteorol. Soc. 2002, 83, 241–248.*

*[5] Jarvis, A.J., Young, P.C., Leedal, D.T., Chotai, A. A robust sequential CO2 emissions strategy based on optimal control of atmospheric CO2 concentrations. Climatic Change, 2008, 86, 357 373.*

*[6] Jarvis, A.J., Leedal, D.T., Taylor, C.J., Young, P.C. Stabilizing global mean surface temperature: a feedback control perspective. Environmental Modelling Software, 2009, 24, 665-674.*

*[7] Ban-Weiss, G.A., Caldeira, K. Geoengineering as an optimization problem. Environ. Res. Lett. 2010, 5, 034009.*

*[8] Weller, S.R.; Schultz, B.P. Geoengineering via solar radiation management as a feedback control problem: Controller design for disturbance rejection. In Proceedings of the 4th Australian Control Conference (AUCC), Canberra, Australia, 17–18 November 2014.*

*[9] Soldatenko S.A. Estimating the impact of artificially injected stratospheric aerosols on the global mean surface temperature in the 21th century. Climate, 2018, 6, 85. doi:10.3390/cli6040085*

*[10] Sierra C.A., Metzler H., Müller M., Kaiser E. Closed-loop and congestion control of the global carbon-climate system. Climatic Change, 2021, 165, 15.*

*[11] Soldatenko, S. and Yusupov, R. An Optimal Control Perspective on Weather and Climate Modification. Mathematics, 2021, 9, 305.*

*[12] Wang Q. Forward and adjoint sensitivity computation for chaotic dynamical systems. Journal of Computational Physics, 2013, 235, 1–13.*

*[13] Soldatenko S.A., Chichkine D. Climate model sensitivity with respect to parameters and external forcing. In: Topics in Climate Modeling. T. Hromadka and P. Rao (eds.), InTech, Rijeka, Croatia, 2016, p. 105-135.*

*[14] Pena M., Kalnay. E. Separating fast and slow modes in coupled chaotic systems, Nonlinear Processes in Geophysics, 2014, 11, 319–327.*

[15] Siqueira L., Kirtman B. *Predictability of a low-order interactive ensemble. Nonlinear Processes in Geophysics, 2012, 19, 273–282.*

**Response:** We thank the reviewer for providing this comprehensive list of references. Relevant citations have been integrated into the revised manuscript to improve context and background.

**Response to Anonymous Referee 2**

1. *This article aims to benchmark the Control Simulation Experiment (CSE) framework for controlling chaotic systems proposed by Miyoshi and Sun (2022) with respect to a well-known method of control theory known as Model Predictive Control (MPC). The study uses the Lorenz 63 (L63) model to perform this analysis, with the wings of the butterfly-like attractor defined as "weather regime". The studied control algorithms must prevent the switch from one wing to the other within an Observing Systems Simulation Experiment (OSSE) framework.*

   **Response:** We thank the referee for summarizing the nature of our work. Our study benchmarks the CSE framework proposed by Miyoshi and Sun (2022) against MPC, using the Lorenz 63 model with the attractor's wings interpreted as "weather regimes" to prevent regime switches.

2. ***General comment***

   ===============

   *First, the paper lies on the conceptual side of the field and cites correctly the related literature, which would have been sufficient for an article on the control of chaotic systems alone. However, here the authors introduction shows that they have weather regimes shift in mind and frame it as a step "[guiding] future research efforts aimed at mitigating the impact of extreme weather events through controllability" while not providing any context or link to the literature of such events controllability. For example, the problem of defining what a weather regime is, is in itself an open problem, and some people might find it difficult to equate weather regimes to the wings of the L63 attractor.*

   *But this first point could be easily addressed by rewriting (substantially) the introduction and conclusion of the paper.*

   **Response:** We agree with the reviewer that our original introduction did not sufficiently provide context or links to the literature regarding the controllability of weather regimes. In particular, we acknowledge that defining a weather regime remains an open problem, and that equating weather regimes to the wings of the Lorenz 63 attractor is a conceptual simplification used for our analysis.

   **Revised Parts:** In the revised manuscript, we have rewritten both the Introduction and Conclusion to explicitly address these issues and to incorporate additional relevant literature.

3. *However, more important problems might lie in the conception of the benchmark itself.*

*The methodology of the benchmark must be detailed more thoroughly and this is why I will suggest a major revision, with a subsequent revision (from me, but also ideally from other referees).*

*I will now detail this:*

***CSE methodology***

*——————————-*

*The problems start with the description of the CSE framework. In Miyoshi and Sun, they (rightfully) take a great deal of precaution stating that "It is essential that our prediction and control system is blind to the NR and takes only the imperfect observations." .*

*The NR is the nature run, and within the OSSE framework, it is a synthetic independent run of the model which is supposed to represent the "truth" (for example the true atmospheric state). However, the authors here never mention the existence of this crucial run in Section 2. When CSE impacts the NR (by perturbing the true atmosphere), it is supposed to represent an actual action on reality, yet the authors state in the introduction (line 24) that "By applying infinitesimal perturbations to the atmospheric state within numerical models, CSE aims to influence the future evolution of chaotic systems toward more desirable trajectories." which hints at a confusion of the authors between a pure model world and a realistic framework containing a natural state to which the perturbations must be applied.*

*The CSE algorithms presentation detailed in section 2 is not wrong in itself, but hides how the observations are generated.*

*The point 4.(d) simply states that perturbations should be added to the "Lorenz 63 model at each time step [between two subsequent analysis]" , while actually in real condition, perturbations would be done on the real atmosphere (represented here by the NR).*

*This is not a problematic description, just an incomplete one of the CSE algorithm. But I think it might lead later to an incorrect comparison with the MPC.*

**Response:** We acknowledge the referee's point that the description of the CSE framework in our manuscript was not clear. In the revised manuscript, we have clarified the explanation of the CSE, specifically addressing the role of the nature run in the methodology and how the observations are generated.

**Revised Parts:** These revisions are made in both the Introduction and Section 2.

4. ***MPC methodology***

*——————————-*

*MPC applies continuously (i.e. at any time step) perturbations over the whole prediction horizon T. These perturbations are obtained by optimizing (e.g. by gradient descent) the discrete-time forecast model evolution considered as a feed-forward network.*

*One could question the interest or feasibility to apply constantly changes to the atmosphere, whether or not an extreme event is forecasted.*

*But that being put aside, it is not clear from the manuscript how the DA is done for the MPC, or if there is a DA process actually involved. In section 3.3, it is said that at each time step the current state of the system is "observed or estimated". Which system? And what does "observed or estimated" here mean? In the CSE framework, it was clear that observations were obtained by sampling and perturbing the NR, but here it is not clear what is used as observations. It looks also like the observations take place at every time step, which is unrealistic for atmospheric forecasting systems.*

*Section 4.2 indicates that the authors tested the 40 initial conditions (IC) provided by Miyoshi and Sun. For instance, in Miyoshi and Sun, the 40 IC are used to generate independent runs of 8000 timeunits with 1000 DA cycles each (i.e. DA takes place each $T_a = 8$ timeunits). Do the runs used for the MPC have the same total length? Or the same at least the same number of observation/assimilation cycles? This is important to see if both statistics are comparable.*

*But the nature of the experiments seems also to be different from the CSE ones.*

*Indeed, it is difficult to know from the manuscript on what these MPC perturbations are applied? On a NR? But how was this NR defined?*

*On the other hand, if there is no NR, then this means that the MPC experiments are not OSSE ones. The authors are then just trying to control runs of a chaotic system (here to the L63, but it could have been any other), forcing its trajectory to satisfy some bounds during a given period. These experiments cannot be linked to any real world system then.*

*Because of all of this, both experiments might not be comparable.*

*Therefore, the authors should detail thoroughly the MPC experiment with regards to the points raised above.*

*Until then, it is not clear if the present study is acceptable for publication.*

**Response:** We agree with the referee that our description of the MPC control algorithm was not clear. In the revised manuscript, we have updated our description to explicitly clarify the following points:

(a) Observation Process: The process of obtaining observations from the NR in the MPC experiments is identical to that used by Miyoshi and Sun, meaning that the noise added (both in mean and variance) is the same as in their work.

(b) Data Assimilation (DA): No data assimilation process is involved in the MPC control algorithm itself; instead, the control optimization uses the directly observed (noisy) NR values. In contrast, the CSE framework employs DA to obtain an analysis state. By omitting DA in our MPC experiments, we evaluate MPC under less favorable conditions, thereby challenging the method to perform without the enhanced state accuracy provided by DA.

(c) Run Length and Assimilation Cycles: The total length of NRs and the number of observation/assimilation cycles in the MPC experiments are the same as those used in the CSE experiments, which ensures comparability between the two methodologies.

(d) Application of Perturbations: The MPC perturbations are indeed applied to the NR. We now clearly state how the NR is defined and how perturbations are introduced into it within our MPC framework.

To address these points, we have thoroughly updated the description of our control algorithm in Section 3.3. We believe that these revisions clarify the MPC methodology and demonstrate that our experiments are directly comparable to the CSE framework as implemented by Miyoshi and Sun.

**Revised Parts:** The control algorithm in Section 3.3 has been rewritten.

5. ***How to improve the manuscript***

   ==============================

   *As said above, the MPC experiments must be better described, in particular how the observations are obtained. This is a crucial point.*

   **Response:** We agree with the referee that our description of the observation process in the MPC experiments requires further clarification. Accordingly, in the revised manuscript, we have updated the description of control algorithms Section 2.2 and 3.3 to state how the observations are obtained from the Nature Run, ensuring consistency with the procedure used by Miyoshi and Sun.

   **Revised Parts:** The control algorithm in Section 3.3 has been rewritten.

   *" An observation is obtained by adding Gaussian noise to the NR ... "*

6. *The MPC with temporal deep unfolding must be better introduced in general, with schematics, so that the reader can understand what it is about, what architecture it is. Citing Kishida and Ogura 2022 is not enough and the readers of the geophysics community deserve a proper introduction to this.*

   **Response:** We agree with the referee that our original manuscript did not sufficiently introduce MPC with temporal deep unfolding. In the revised manuscript, we have significantly revised Section 3.3 to provide a clear description of the architecture, including schematics, so that readers, especially those from the geophysics community, can readily understand the method.

   **Revised Parts:** Section 3.3 has been updated. The MPC with temporal deep unfolding algorithm has been rewritten on Page 9 of the revised manuscript. Figure 1, which provides a conceptual framework of Model Predictive Control via Temporal Deep Unfolding, has been added to the revised manuscript.

7. *Also the MPCIL is mentioned first in section 4.2, without any prior explanation, so that readers not reading Kishida and Ogura cannot understand what this is. MPCIL must be properly introduced beforehand.*

   **Response:** We agree with the referee that MPCIL was not clearly introduced in the original manuscript. Incremental Learning (IL) is a machine learning technique that enables a model to continuously update itself with new data without requiring full retraining. This allows the model to adapt to evolving system dynamics while retaining past knowledge. In

the context of Model Predictive Control via Temporal Deep Unfolding (MPC-TDU), IL plays a crucial role in maintaining an adaptive predictive model over time. Instead of recomputing the optimization problem from scratch at each step, IL incrementally refines previously learned models, enhancing computational efficiency and adaptability to dynamic system changes. The corresponding algorithm is referred to as MPCIL.

**Revised Parts:** In the revised manuscript, we have added a detailed explanation of MPCIL in line 228, line 265, respectively, to ensure that readers unfamiliar with Kishida and Ogura's work can fully understand its concepts and implementation.

*" In the context of temporal deep unfolding, incremental learning is a technique where the model learns from new data incrementally without retraining from scratch, preserving knowledge from previous learning ... "*

*" When incremental learning is applied, Step 2c becomes as follows ... "*

8. *Final words*

    ===========

    *Also the code of the study must be shared openly, as it is strongly recommended in NPG.*

    *It should be noted that not releasing the code for studies using the simple L63 model defeats the point of using L63, i.e. providing a conceptual framework where reproducing experiments is easy.*

    **Response:** We thank the referee for this thoughtful and constructive suggestion. In addition to revising the manuscript accordingly, we have now made all code used in our experiments publicly available at https://github.com/ogura-lab/NPG_CSE-MPC. We believe that releasing our code will facilitate reproducibility and further exploration of our conceptual framework using the Lorenz-63 model.

Please do not hesitate to contact us for further information.

Sincerely,

Rikuto Nagai, Yang Bai, Masaki Ogura, Shunji Kotsuki, and Naoki Wakamiya

---

## Editor Decision (ED1)

The two referees are the same as for the first version of the paper, with the same identification numbers.

Referee #1 writes that the authors have revised the paper taking into account his/her comments, and that the paper can be published as such.

Referee #2 is more critical. His/her criticisms do not bear on the science of the paper, but on what he/she considers as *unsubstantiated claims*, that need to be clarified. He/she gives specific examples.

I ask the authors to revise their paper along the lines suggested by Referee #2, and to give a point-by-point response to each of those comments.

I also as Editor have a number of specific suggestions for modification (contrary to what is the case in Referee #2's comments, the line numbers below are those of the file npg-2024-26-ATC1.pdf, which shows explicitly the modifications made by the authors in their paper).

1. L. 137, (a), select only two ensemble members $S$ and $N$ ? And what if all elements predict a regime shift ? Go to the next time step, although action obviously seems to be required ?

2. L. 248-250, Minimizing cost function (6) with condition (7) will tend to make the values of $x_t$ negative, although it is positive values that are looked for (see ll. 314-317).

3. L. 302, *Figure 3.a → Figure 4.a*
4. L. 303, *Figure 3.b → Figure 4.b*
The same correction is to made later in the paper. Please check.

5. Ll. 321-322, … *the control success rate with incremental learning was never lower than without incremental learning for any combination of parameters T and D*

   And then (l. 328), *This result indicates that MPCIL outperforms MPC in some cases …* Only in some cases, or all (see also l. 332, … *in certain scenarios*.) ?

In case the authors disagree with a particular comment or decide not to follow a particular suggestion (whether from the Referee or the Editor), they must state precisely their reasons for that.

I will be looking forward to a new revised version of the paper, which I may submit to further review.

---

## Editor Decision (ED2)

I thank the authors for this new version of their paper. I consider it can now be accepted from the scientific point of view. But there remain a number of points to be improved as concerns the edition of the paper. Here are examples (the line numbers are those of file Authors_tracked_changes.pdf)

1. In response to a comment by Referee #2, the authors now mention how the Nature Run has been produced (ll. 123-124). However, that is done as a passing remark when describing how 'observations' have been obtained. And the specific notation of the Run ($x_t^{NR}$) is introduced only on l. 230.
I think all information on the Nature Run should be introduced in the first place it is mentioned (ll. 115-116).

2. Ll. 252-253, *Miyoshi and Sun* […] *conducted experiments* […] *for 40 different initial conditions*. Did those 40 different initial conditions correspond to 40 different Nature Runs, or what ?

3. Similarly, ll. 270-271, *The initial values of the system were set …* This now refers to experiments performed with MPC. I understand that these initial values were not those of the Nature Run. But what were they exactly ? How do they differ from the 40 initial values of Miyoshi and Sun (which are again used for MPC, as mentioned on l. 285)

4. The value of the observation period $T_a$ (which I understand is always $T_a = 8$ steps) is not mentioned in Table 1.

5. L. 232, … *the state transition from time t + k to time t + k + 1*

These are only examples. I suggest that the authors check carefully their paper for possible inconsistencies, ambiguities or inaccuracies.